# A decade of stability for *w*Mel *Wolbachia* in natural *Aedes aegypti* populations

**Perran A. Ross**[1]*, **Katie L. Robinson**[1], **Qiong Yang**[1], **Ashley G. Callahan**[1], **Thomas L. Schmidt**[1], **Jason K. Axford**[1], **Marianne P. Coquilleau**[1], **Kyran M. Staunton**[2], **Michael Townsend**[2], **Scott A. Ritchie**[2], **Meng-Jia Lau**[1], **Xinyue Gu**[1], **Ary A. Hoffmann**[1]

1 Pest and Environmental Adaptation Research Group, Bio21 Institute and the School of BioSciences, The University of Melbourne, Parkville, Victoria, Australia, 2 Australian Institute of Tropical Health and Medicine, James Cook University, Smithfield, Queensland, Australia

* perran.ross@unimelb.edu.au

## Abstract

Mosquitoes carrying *Wolbachia* endosymbionts are being released in many countries for arbovirus control. The *w*Mel strain of *Wolbachia* blocks *Aedes*-borne virus transmission and can spread throughout mosquito populations by inducing cytoplasmic incompatibility. *Aedes aegypti* mosquitoes carrying *w*Mel were first released into the field in Cairns, Australia, over a decade ago, and with wider releases have resulted in the near elimination of local dengue transmission. The long-term stability of *Wolbachia* effects is critical for ongoing disease suppression, requiring tracking of phenotypic and genomic changes in *Wolbachia* infections following releases. We used a combination of field surveys, phenotypic assessments, and *Wolbachia* genome sequencing to show that *w*Mel has remained stable in its effects for up to a decade in Australian *Ae. aegypti* populations. Phenotypic comparisons of *w*Mel-infected and uninfected mosquitoes from near-field and long-term laboratory populations suggest limited changes in the effects of *w*Mel on mosquito fitness. Treating mosquitoes with antibiotics used to cure the *w*Mel infection had limited effects on fitness in the next generation, supporting the use of tetracycline for generating uninfected mosquitoes without off-target effects. *w*Mel has a temporally stable within-host density and continues to induce complete cytoplasmic incompatibility. A comparison of *w*Mel genomes from pre-release (2010) and nine years post-release (2020) populations show few genomic differences and little divergence between release locations, consistent with the lack of phenotypic changes. These results indicate that releases of *Wolbachia*-infected mosquitoes for population replacement are likely to be effective for many years, but ongoing monitoring remains important to track potential evolutionary changes.

## Author summary

*Wolbachia* are endosymbionts that can block the transmission of arboviruses by mosquitoes. *Aedes aegypti* mosquitoes carrying the *w*Mel strain of *Wolbachia* have been released in 'population replacement' interventions, which aim to establish *w*Mel in mosquito populations, thereby reducing their ability to spread disease. *Wolbachia* population

---

**Data Availability Statement:** All experimental data are within the manuscript and its Supporting Information files. Raw read datasets and wMel

---

genome sequences have been deposited in Genbank under BioProject numbers PRJNA776956 and PRJNA791959.

**Funding:** AH was supported by the National Health and Medical Research Council (1132412, 1118640, www.nhmrc.gov.au). The funders had no role in study design, data collection and analysis, decision to publish, or preparation of the manuscript.

**Competing interests:** The authors have declared that no competing interests exist.

replacement programs began only a decade ago, raising uncertainty about their long-term effectiveness. Here we provide a comprehensive assessment of the long-term stability of *w*Mel from the very first *Wolbachia* population replacement release. We show that there is no evidence for changes in the phenotypic effects of *w*Mel in mosquitoes, and confirm that the *w*Mel genome has changed very little in the decade since field releases began. *w*Mel remains at high levels within mosquitoes, suggesting that its ability to block virus transmission has been retained. Our data provides confidence that *Wolbachia* population replacement releases will provide ongoing protection against arbovirus transmission.

## Introduction

Open field releases of *Wolbachia*-infected mosquitoes are becoming one of the best ways to control arbovirus transmission. *Wolbachia* "population replacement" programs involve the release of mosquitoes carrying a *Wolbachia* infection that spreads through mosquito populations and reduces their vector competence [1–3]. Several different *Wolbachia* strains from other insects have been introduced into *Ae. aegypti* mosquitoes through microinjection, with many of them reducing the ability of mosquitoes to transmit arboviruses including dengue, Zika and chikungunya [4–6]. The establishment of *w*Mel and *w*AlbB *Wolbachia* strains in natural populations has suppressed arbovirus transmission by *Ae. aegypti* in release locations [7,8]. Population replacement relies on maternal transmission of *Wolbachia* as well as cytoplasmic incompatibility between *Wolbachia*-infected males and uninfected females to drive and maintain the *Wolbachia* infection into the population. Successful establishment and ongoing persistence depends on properties of the *Wolbachia* strain as well as local environmental conditions which can influence mosquito dispersal [9], *Wolbachia* maternal transmission [10], cytoplasmic incompatibility [11] and host fitness effects of *Wolbachia* [12,13].

*Wolbachia* releases for population replacement first took place in 2011, where *w*Mel-infected *Ae. aegypti* were released in two suburbs of Cairns, Australia: Gordonvale and Yorkeys Knob [14]. The *w*Mel infection rapidly increased in prevalence and has persisted at a high frequency in these suburbs for many years [15,16]. Releases of *w*Mel-infected *Ae. aegypti* have since expanded to cover nearly the entire distribution of *Ae. aegypti* in Australia [16,17]. Following the stable establishment of *w*Mel in almost all release locations, local dengue transmission has almost been eliminated in the country [16,17].

Releases of *w*Mel-infected *Ae. aegypti* have now been carried out in several dengue-endemic cities including Yogyakarta, Indonesia [18] and Rio de Janeiro [19,20] and Niterói [21,22], Brazil. Quasi-experimental and/or randomized controlled trials show that *w*Mel releases have reduced dengue incidence by >69% [8,22,23], with reductions in chikungunya and Zika transmission also apparent in some locations [22]. Long-term monitoring shows that *w*Mel has persisted in *Ae. aegypti* populations at high frequencies for many years [16,17,21], though in some locations the infection has remained at an intermediate frequency or dropped out, requiring supplemental releases [19,20,22]. *w*Mel frequencies can also fluctuate seasonally, likely due to high temperatures experienced in larval habitats [24,25].

Theory predicts that *Wolbachia* infections, mosquitoes and viruses may evolve, potentially rendering *Wolbachia* population replacement less effective over time [26–28]. Previous studies in *Drosophila* demonstrate the potential for evolutionary changes affecting both the *Wolbachia* and host genomes. The *w*Ri *Wolbachia* strain invaded Australian populations of *Drosophila simulans* and rapidly shifted from inducing a host fitness cost to a host fitness benefit [29]. The *w*MelPop strain also induced weaker host fitness costs and cytoplasmic incompatibility across

time after being transferred to a novel *Drosophila* host [30,31]. Selection experiments show that shifts in the phenotypic effects of *Wolbachia* are often due to host genetic changes [32]. So far there have been limited changes observed in *Wolbachia* genomes across several years after transinfection to novel hosts such as *w*Cer2 in *Drosophila* [33] and *w*MelPop-CLA in *Ae. aegypti* [34].

In mosquitoes, there is a clear distinction between natural *Wolbachia* infections and novel transinfections, where the latter tends to induce deleterious effects [1]. This suggests that the effects of *Wolbachia* transinfections may weaken across evolutionary timescales. Since population replacement programs began only a decade ago, there is still limited information on evolutionary changes following deliberate *Wolbachia* invasions. In laboratory populations of *Ae. aegypti*, the *w*MelPop-PGYP strain has continued to induce complete cytoplasmic incompatibility for the last decade, but some fitness costs appear to have weakened [35]. In field populations, the *w*Mel strain has shown stable phenotypic effects, with no evidence for changes in dengue virus blocking [21,36], cytoplasmic incompatibility [15,37] or effects on fertility [15] after a period of 1 year or more under field conditions. Whole *Ae. aegypti* [38] and *Wolbachia* [39,40] genome sequencing studies show limited genomic changes after *w*Mel has been established in Cairns for at least 7 years.

Given the importance of tracking the long-term stability of *Wolbachia* infections in mosquito populations, we have now collected additional data on the phenotypic and genomic stability of *w*Mel from the first ever population replacement releases in Cairns, Australia in 2011. Our data from up to a decade post-release show that the phenotypic effects of *w*Mel largely remain stable in both laboratory and near-field *Ae. aegypti* populations. We also extend the findings of Huang et al. [39] to show limited genomic changes in *Wolbachia* over the span of a decade, with no divergence in *w*Mel genomes between different release locations. Our data point to the likely long-term effectiveness of *Wolbachia* population replacement programs globally.

## Results

### *w*Mel remains at a high frequency in Cairns following releases

We performed ovitrapping across suburbs in Cairns in 2016 and 2018 to monitor *Wolbachia* infection frequencies in the *Ae. aegypti* population. In 2016, all suburbs where *w*Mel releases had taken place had a *w*Mel infection frequency above 0.96 (S1A Fig). Some pre-release suburbs had low infection frequencies (0.05–0.21), indicating spread of *w*Mel to adjacent suburbs (e.g. to Mt Sheridan and Holloways Beach). By 2018, releases had occurred in additional locations (e.g. Trinity Beach and Redlynch) and all suburbs had wMel infection frequencies greater than 0.93 (S1B Fig), except for Redlynch where *w*Mel was not released until 2019. Our data are broadly consistent with *Wolbachia* infection frequencies from an independent study which also shows that *w*Mel has maintained high frequencies in most release locations [16].

### *w*Mel density remains stable across suburbs and laboratory generations

We measured *w*Mel density in 4th instar larvae reared in the laboratory from ovitraps collected across suburbs in 2018. We found no clear effect of release year (GLM: $F_{4,269} = 2.274$, P = 0.062) or suburb (nested within release year) ($F_{6,269} = 2.017$, P = 0.064) on *Wolbachia* density (Fig 1A). These results suggest that the *w*Mel infection has remained stable after being established in the field for different periods of time (from 1–7 years), with no clear effects of local environmental conditions on *w*Mel in the next generation of mosquitoes.

In June 2019, we measured *w*Mel density in adults from laboratory populations that were established from field collections at different times. We found no significant differences

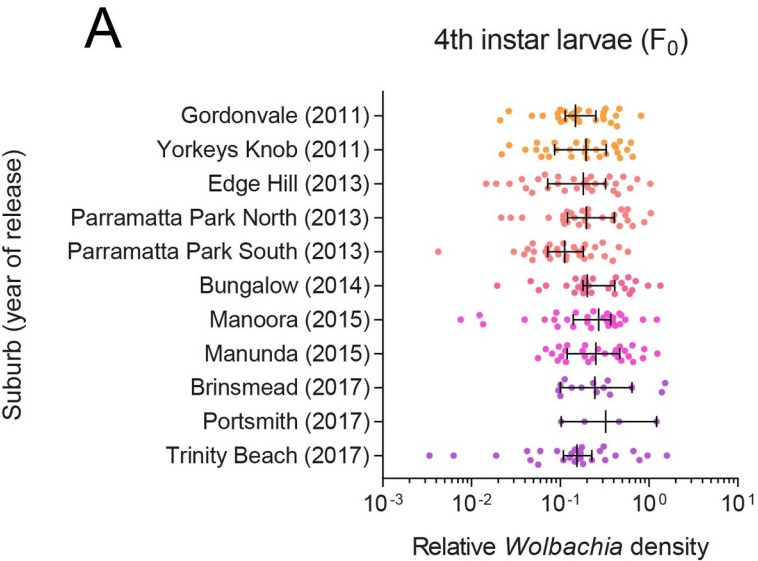

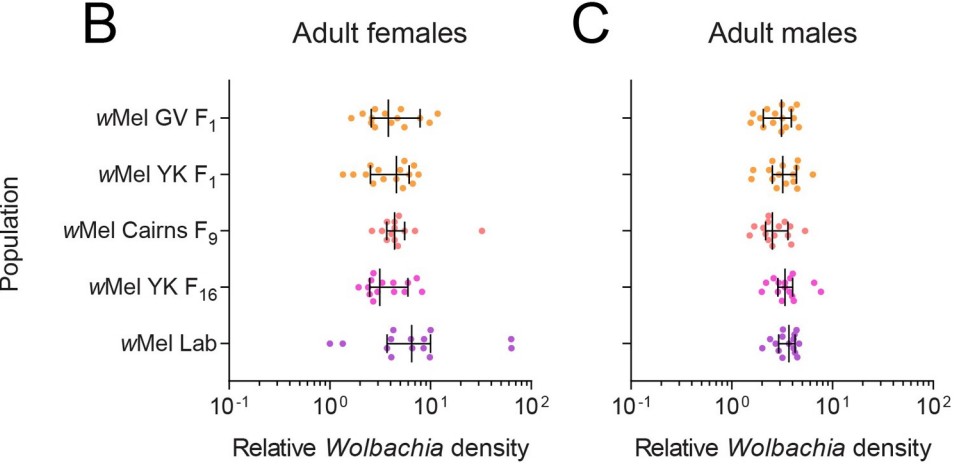

**Fig 1. *w*Mel density remains stable across suburbs and laboratory generations.** (A) *Wolbachia* density in 4th instar larvae hatched from ovitraps collected across Cairns suburbs in February-March 2018. The year where releases were undertaken in each suburb is shown in parentheses. (B-C) *Wolbachia* density in adult (B) females and (C) males in populations that had been reared in the laboratory for different numbers of generations. Dots represent data from three technical replicates of an individual mosquito. Vertical lines and error bars represent medians and 95% confidence intervals.

between populations for females (GLM: $F_{4,65}$ = 1.767, P = 0.146, Fig 1B) or males ($F_{4,70}$ = 1.942, P = 0.113, Fig 1C), suggesting that whole-adult *w*Mel density has not changed across different durations of laboratory rearing.

## Population origin and *w*Mel infection influence mosquito fitness

In September 2018 we performed phenotypic comparisons between long-term laboratory and near-field populations that were *w*Mel-infected or cleared of *w*Mel through antibiotic treatment (Fig 2). We found significant effects following Bonferroni correction (adjusted α: 0.008) of population origin on larval development time (females: $F_{1,44}$ = 19.656, P < 0.001, males:

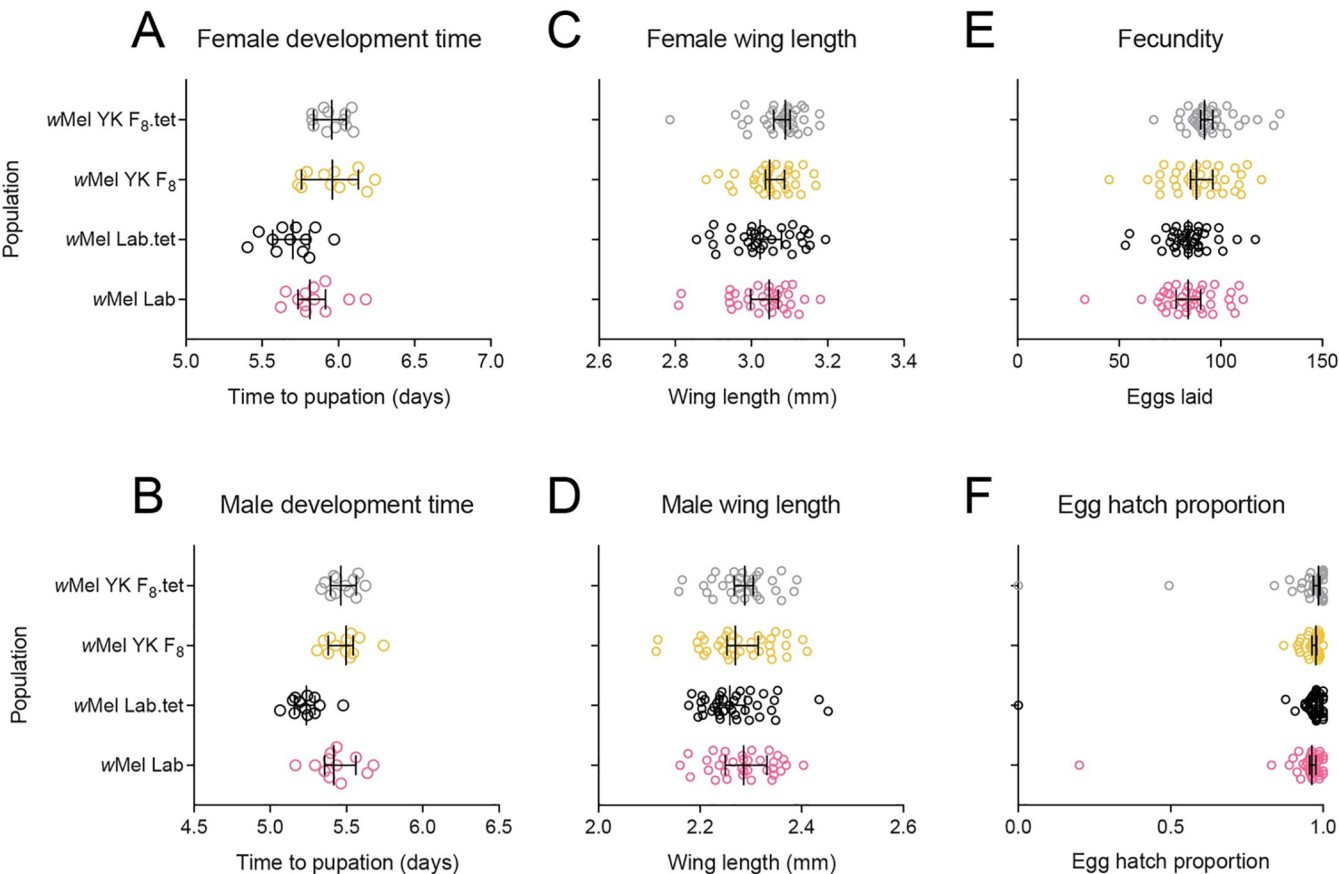

**Fig 2. Phenotypic effects of *w*Mel in laboratory and field *Aedes aegypti* backgrounds.** *w*Mel-infected populations were compared to tetracycline-cured counterparts to control for genetic background between infected and uninfected mosquitoes. Populations were measured for (A) female and (B) male development time, (C) female and (D) male wing length, (E) fecundity and (F) egg hatch. Data from two replicate populations were pooled for visualization. Dots represent data from replicate trays (A-B) or individual mosquitoes (C-F). Vertical lines and error bars represent medians and 95% confidence intervals.

$F_{1,44}$ = 18.266, P < 0.001) and female fecundity ($F_{1,150}$ = 11.640, P = 0.001) but not wing length (females: $F_{1,140}$ = 6.107, P = 0.015, males: $F_{1,141}$ = 0.303, P = 0.583) or (logit transformed) egg hatch proportions ($F_{1,150}$ = 0.305, P = 0.581). *w*Mel infection had no significant effect on any trait (all P > 0.089) except for male development time ($F_{1,44}$ = 9.296, P = 0.004). For this trait, we also found an interaction between *w*Mel infection and population origin ($F_{1,44}$ = 7.958, P = 0.007), where *w*Mel infection increased male development time in the lab populations but not the YK (Yorkeys Knob) populations.

## No clear effect of *w*Mel origin in a common host background

In the first experiment, we found that effects on fitness were driven mainly by population origin. Due to potential interactions between *Wolbachia* infection and background in the first experiment, we performed a second set of experiments in August 2020 to evaluate the effects of lab and field-derived *w*Mel infections in a common host background. We found no significant differences following Bonferroni correction (adjusted α: 0.006) between populations for most traits tested, including development time (GLM: females: $F_{3,43}$ = 2.841, P = 0.049, males: $F_{3,43}$ = 3.558, P = 0.022), female wing length ($F_{3,74}$ = 1.826, P = 0.150), fecundity ($F_{3,433}$ = 1.950, P = 0.121) and adult longevity (Log-rank: females: $\chi^2$ = 5.700, P = 0.127, males: $\chi^2$ =

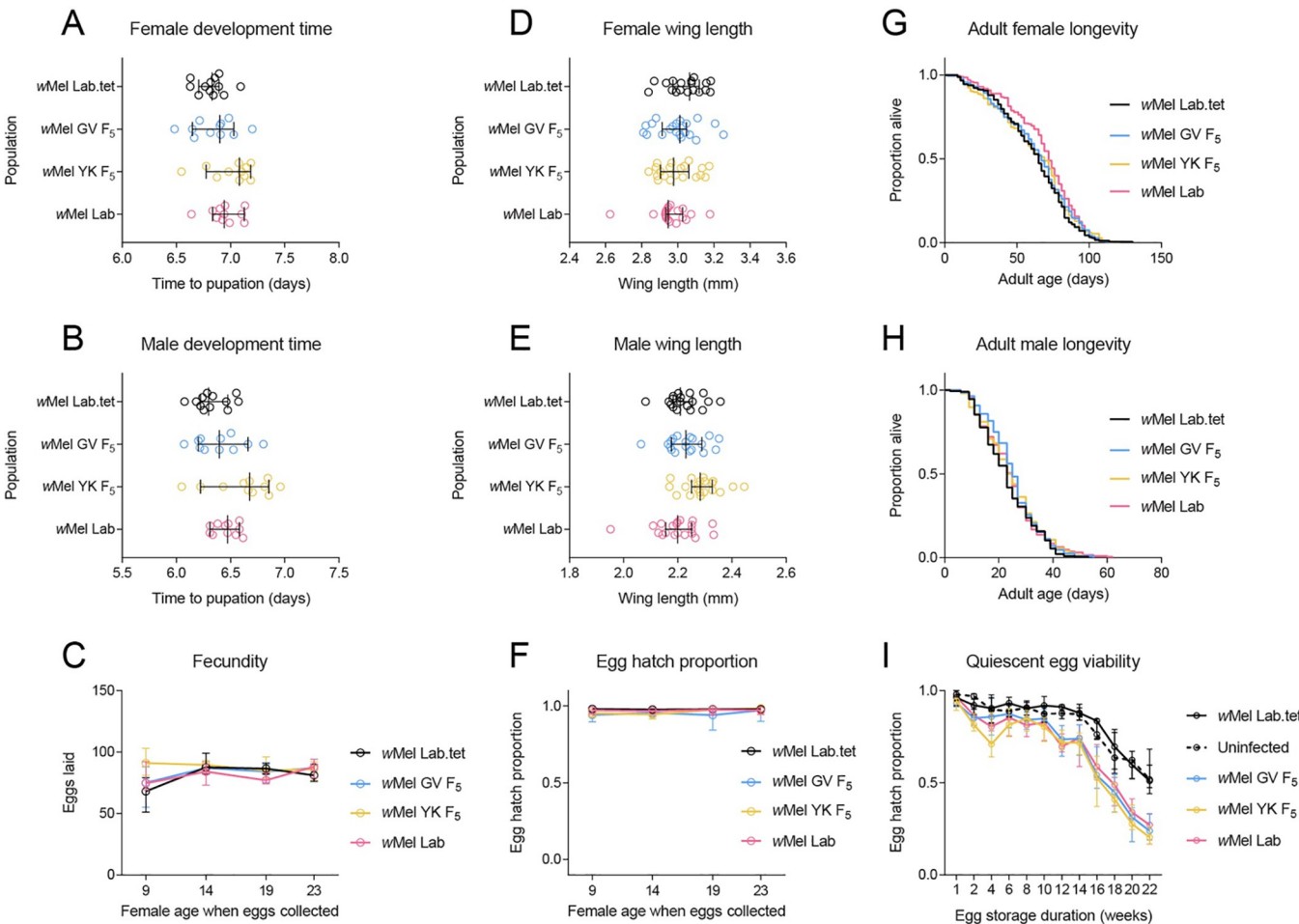

**Fig 3. Phenotypic effects of *w*Mel from lab and field origins in a common *Aedes aegypti* background.** *w*Mel-infections originating from Gordonvale (GV), Yorkeys Knob (YK) or the laboratory were introduced to a common background through backcrossing. Populations were measured for (A) female and (B) male development time, (C) fecundity, (D) female and (E) male wing length, (F) egg hatch, (G) female and (H) male adult longevity and (I) quiescent egg viability. Dots represent data from replicate trays (A-B) or individual mosquitoes (D-E). Lines and error bars represent medians and 95% confidence intervals in all panels.

3.428, P = 0.330, Fig 3). There was also no significant effect of population on fecundity and (logit transformed) egg hatch (all P > 0.056), except for fecundity in gonotrophic cycle 1 ($F_{3,107}$ = 5.847, P = 0.001), where *w*Mel YK females laid more eggs (Fig 3C), and (logit transformed) hatch proportion in gonotrophic cycle 3 ($F_{3,108}$ = 5.449, P = 0.002), with *w*Mel GV having reduced egg hatch compared to the other populations (Fig 3F). Population effects were also observed for male wing length (GLM: $F_{3,73}$ = 5.293, P = 0.002), with *w*Mel YK males having larger wings (Fig 3E). In the quiescent egg viability experiment, we found no significant effect of population on (logit transformed) hatch proportion after 1 week (GLM: $F_{4,55}$ = 1.110, P = 0.361). Egg viability declined more rapidly for *w*Mel-infected populations (Fig 3I), with substantial differences between populations by week 22 ($F_{4,47}$ = 35.563, P < 0.001). However, there were no significant differences among *w*Mel-infected populations at the same time point ($F_{2,27}$ = 1.156, P = 0.330). Overall, our results demonstrate few consistent and strong effects of *w*Mel origin on mosquito fitness.

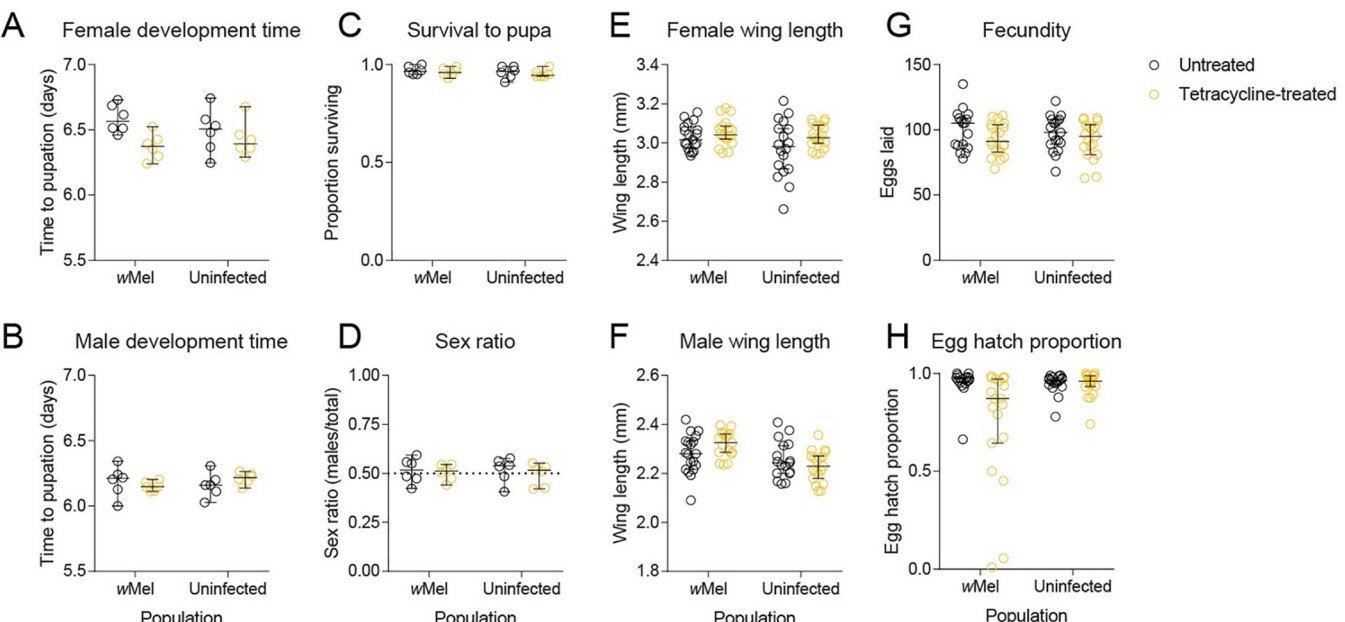

**Fig 4. Phenotypic effects of parental tetracycline treatment in *Aedes aegypti*.** *w*Mel-infected mosquitoes or uninfected mosquitoes were left untreated or fed 2 mg/mL tetracycline. Their offspring were measured for (A) female and (B) male development time, (C) survival to pupa, (D) pupal sex ratio, (E) female and (F) male wing length, (G) fecundity and (H) egg hatch. Dots represent data from replicate trays (A-D) or individual mosquitoes (E-H). Lines and error bars represent medians and 95% confidence intervals in all panels.

## Limited effects of antibiotic treatment on fitness

To test whether tetracycline treatment could influence mosquito fitness, we performed phenotypic comparisons of *w*Mel-infected and uninfected mosquitoes following parental tetracycline treatment (Fig 4). We found no significant effect of tetracycline treatment on any trait (all P > 0.007, adjusted α: 0.006) except for (logit transformed) egg hatch proportion where we found an interaction between treatment and *Wolbachia* infection type ($F_{1,70}$ = 9.261, P = 0.003). Decreased egg hatch due to tetracycline treatment was apparent only for the *w*Mel-infected population, likely due to incomplete curing resulting in partial self-incompatibility. In this experiment, we also found no significant effect of *Wolbachia* infection type for any trait (all P > 0.017, adjusted α: 0.006) except for male wing length, where *w*Mel-infected males had larger wings ($F_{1,69}$ = 12.387, P = 0.001). These results provide confidence that observed effects in comparisons between *Wolbachia*-infected and cured mosquitoes are due to removal of the *Wolbachia* infection and not off-target tetracycline-related effects on the gut microbiome or mitochondria.

## *w*Mel induces complete cytoplasmic incompatibility after eight years in the field

In June 2019, we tested the ability of *w*Mel-infected males from different origins (Yorkeys Knob, Gordonvale and laboratory) to induce cytoplasmic incompatibility with uninfected females. Males from all *w*Mel-infected populations induced complete cytoplasmic incompatibility, with no eggs hatching in crosses with uninfected females (Table 1). All other crosses yielded egg hatch proportions above 90% (Table 1), consistent with data from the previous experiments showing limited effects of *w*Mel infection on egg hatch. These results show that *w*Mel has retained complete cytoplasmic incompatibility after > 8 years in the field and one generation of laboratory rearing.

**Table 1. Reciprocal crosses between *w*Mel-infected *Aedes aegypti* from laboratory or field origins and uninfected *Ae. aegypti*.** Crosses with "-"were not tested.

| | | Median egg hatch proportion (lower, upper 95% confidence interval) | | | |
|---|---|---|---|---|---|
| | | Female | | | |
| | | Uninfected | *w*Mel Lab | *w*Mel YK $F_1$ | *w*Mel GV $F_1$ |
| Male | Uninfected | 0.946 (0.922, 0.971) | 0.960 (0.915, 0.981) | 0.958 (0.800, 0.973) | 0.974 (0.964, 0.981) |
| | *w*Mel Lab | 0 (0, 0) | 0.959 (0.949, 0.978) | - | - |
| | *w*Mel YK $F_1$ | 0 (0, 0) | - | 0.973 (0.924, 0.991) | - |
| | *w*Mel GV $F_1$ | 0 (0, 0) | - | - | 0.974 (0.958, 0.991) |

## Limited changes in *w*Mel and mitochondrial genomes across a decade in a novel *Aedes aegypti* host

Previous *w*Mel genome comparisons suggest limited changes in the *Wolbachia* genome have occurred since field releases [39,40]. We extend this timeframe by including pre-release material from 2010 and field collections from Gordonvale and Yorkeys Knob in 2020. We also include a dataset from a previous whole genome sequencing study of individuals collected from Gordonvale in 2013 and 2018 and Yorkey's Knob in 2018 [38]. Little evidence of change was observed among the analysed *w*Mel genomes, for which we achieved $\geq$ 99.99% coverage. Within populations, each of which was represented by a sample of pooled individuals, seven loci were found to have one or more alternate alleles present at a frequency greater than our threshold value of 25% in at least one sample, all of which were SNPs or small indels. At all other positions within the genome, the *w*C45 $F_{10}$ (pre-release) consensus sequence was identical to that of the Gordonvale and Yorkeys Knob field collections from 2011, 2018 and 2020, and the *w*Mel lab population (see also Huang et al. [39] and Dainty et al. [40]). Relative to these genomes, one single nucleotide deletion was observed in the *w*C45 $F_9$ pre-release genome sequence and seven single nucleotide indels were detected in the 2013 Gordonvale field collection genome sequence (S1 Table). It is probable that these eight indels represent sequencing artifacts, as they involve non-parsimonious changes in populations sampled over multiple time points, were often associated with a marked localized decrease in read depth in the variant samples and mostly represent frameshift mutations.

Of the seven loci that displayed within-sample polymorphism, two were within non-coding regions; three were within three different copies of the group II intron-encoded protein gene LtrA, each at the same position within the gene and corresponding to an Asp > Asn substitution; one was within a gene encoding a hypothetical protein and corresponds to a silent change; and one was within a tRNA-Arg gene (S2 Table). It is possible that the apparent polymorphism within the LtrA genes is due to sequence variation between gene copies, rather than the presence of alternate alleles. Multiple alleles were observed in all samples for these three loci and for the polymorphic locus within the tRNA-Arg gene. At the other three loci, allele frequencies were more variable, with only some samples displaying polymorphism, most of which were from the field populations at Gordonvale and Yorkeys Knob. Two of these loci, at positions 587,862 and 1,174,712 relative to the *w*Mel reference genome, were previously noted by Huang et al. [39] and Dainty et al. [40]. No clear pattern was observed across different populations or time points. While it is possible that this variation is attributable to underlying fluctuation in allele frequencies within populations, or methodological differences between studies, it is more likely to be due to stochastic sampling effects, given the relatively small number of individuals pooled in some samples.

Similar to the *w*Mel genomes, we detected very few changes between the mitochondrial genomes sequenced in this study, for which we obtained $\geq$ 99.31% coverage. The genomes

shared 100% sequence identity across all gene-encoding regions. Nucleotide differences were observed among the consensus sequences at 24 positions within an approximately 1.8Kb region of the genome that does not contain any predicted gene sequences, is very AT rich and displayed greatly reduced read mapping depths (S1 Table). It is therefore likely that many or all of these differences represent sequencing errors. A further 16 positions within this region were found to show within-sample polymorphism, with alternate alleles present at a frequency greater than the threshold value in at least one sample (S2 Table). The lack of variation observed among the *w*Mel genomes and among the mitochondrial genomes is consistent with a high fidelity of maternal co-transmission of both *Wolbachia* and mitochondria.

## Discussion

In this study, we provide a comprehensive update on the genomic and phenotypic stability of the *w*Mel *Wolbachia* infection, up to ten years after being released in *Ae. aegypti* populations. We observed few changes in the *w*Mel genome since before releases began, with little divergence between different locations despite nearly a decade of separation. Furthermore, *w*Mel retains complete cytoplasmic incompatibility, a stable within-host density, and limited host fitness costs. Maintenance of these phenotypic effects along with virus blocking [21,36] and maternal transmission [15] is crucial for the ongoing persistence of *Wolbachia* and suppression of arbovirus transmission. Our results add to a growing body of evidence (e.g. [15,16,39]) supporting the notion that *w*Mel will continue to remain an effective tool for dengue control for many years, although we note that unlike in the Cairns region of North Queensland, *w*Mel has not always successfully invaded to a high frequency and persisted in some other areas [20,25].

Our results suggest that any shifts in the host phenotypic effects of *w*Mel are likely to reflect evolutionary changes within mosquito populations rather than the *Wolbachia* genome. This hypothesis is consistent with crossing and selection experiments showing that *Wolbachia* phenotypic effects depend on nuclear background [13,41]. It is also consistent with genomic data from this and previous studies showing the long-term stability of *Wolbachia* genomes following transinfection [33,34,39,40]. Our fitness experiments do not provide evidence for evolutionary changes in response to *Wolbachia* infection, with similar effects observed in mosquito populations that had been infected with *w*Mel for many years (experiment 1) and in populations where *w*Mel was recently introgressed (experiment 2). Comparisons of mosquito genomes prior to *w*Mel release in Gordonvale and seven years post-release point to limited changes [38], but further work with replicated populations is required to confirm any changes due to *Wolbachia* infection.

Accurate phenotyping of *Wolbachia* infections requires careful control of the nuclear and mitochondrial background between *Wolbachia*-infected and uninfected populations. In this study, we used tetracycline curing followed by backcrossing to ensure that *Wolbachia*-infected and uninfected populations had matched mitochondrial and nuclear genomes. Cross-generational effects of tetracycline treatment on fitness have been speculated [42], with some studies accounting for potential disruptions to the microbiome by rearing treated mosquitoes in water from untreated mosquitoes (e.g. [43]). We found no evidence to suggest that tetracycline treatment influences fitness in the following generation, at least for a concentration commonly used to cure *Wolbachia* infections in insects [44]. While the effect of tetracycline treatment on the mosquito microbiome (aside from *Wolbachia*) remains to be tested, repeated backcrossing to a common background should help to minimize any differences between populations.

Our data, spanning up to a decade since the first releases of *w*Mel in Australia, should be informative for *Wolbachia* release programs that have taken place more recently in other countries. Our results suggest that when the *w*Mel infection is maintained in an *Ae. aegypti* population, the phenotypic effects associated with *w*Mel invasion are likely to persist given that the infection

remains at a high density. However, more work is required to understand the extent of genetic changes in mosquito populations in response to _Wolbachia_ releases. These may include host genomic changes in response to the presence of _Wolbachia_ or indirect effects from the introduction of host genes from release stocks, such as the introduction of pesticide susceptibility genes in Tubiacanga, Brazil, which likely contributed to an unsuccessful _Wolbachia_ release there [19]. Pesticide susceptibility is an issue in most countries where dengue is endemic and chemicals including pyrethroids, organophosphates, and insect growth regulators are applied to suppress _Ae. aegypti_ populations [45]. However, this has not been an issue in the Cairns region where limited applications of pesticides have likely prevented the local evolution of resistance [46].

While _w_Mel has retained complete cytoplasmic incompatibility and maternal transmission under laboratory conditions, these parameters are affected by environmental conditions [10] and require further evaluation under field conditions. The persistence of a high incidence of _w_Mel in natural populations may be constrained by environmental conditions including high temperatures in some locations [25]. Furthermore, interactions with mosquito genetic background may influence the effects of _Wolbachia_ infection on host fitness [41], which could help to explain variability in _w_Mel establishment success in different countries. While several studies have measured _w_Mel effects in local mosquito backgrounds (e.g. [47,48]), direct comparisons across multiple mosquito strains are needed to understand the contribution of host genetics to phenotype. Finally, ongoing monitoring remains important to identify any changes in _Wolbachia_ infection frequency and inform the need for supplementary releases including those with different host genetic backgrounds and different _Wolbachia_ strains.

## Materials and methods

### Ethics statement

Blood feeding of female mosquitoes on human volunteers for this research was approved by the University of Melbourne Human Ethics Committee (approval 0723847). All adult subjects provided informed written consent (no children were involved).

### Field collections and colony establishment

_Aedes aegypti_ mosquitoes were collected as eggs from suburban Cairns in 2016, 2018, 2019 and 2020 from ovitraps. Felt strips from ovitraps were collected and processed identically to previously described methods [24]. Ovitrapping was performed throughout suburban Cairns in September-October 2016 (19 suburbs, 50–100 traps per suburb) and February-March 2018 (12 suburbs, 40–100 traps per suburb), many of which were targeted by _w_Mel release programs from 2011 to 2017 [14,16,49]. The 2019 and 2020 field collections targeted two suburbs with 20 ovitraps each, Gordonvale and Yorkeys Knob, where _w_Mel-infected _Ae. aegypti_ were released in 2011 [14]. Subsets of _Ae. aegypti_ larvae from central Cairns, Gordonvale and Yorkeys Knob were pooled from all traps within a suburb to establish laboratory populations for phenotypic comparisons. Thirty individuals from the $F_1$ and $F_2$ generations were screened for _Wolbachia_ infection (see below) to confirm fixation of _w_Mel within the laboratory populations. All populations were maintained at census size of ~450 individuals per generation at 26˚C and a 12:12 light:dark cycle as described previously [50]. Female mosquitoes were fed on the forearm of a single human volunteer for egg production.

### _Wolbachia_ infection frequency and density

_Wolbachia_ infection frequencies were estimated from the 2016 and 2018 field collections by screening subsets of individuals hatching from ovitraps for _Wolbachia_. For the 2016 field

collections, life stage was uncontrolled, with a mix of adults and larvae tested, thus data were only suitable for *Wolbachia* frequency estimates. Up to 10 individuals were screened per trap, with between 10 and 243 individuals screened per suburb. For the 2018 field collections, 30 4$^{th}$-instar larvae from 15 ovitraps (2 per trap) per suburb were screened for *Wolbachia* infection. Larvae were reared at a controlled density (50 larvae per tray) and stored at the same age (5 d post-hatching), allowing for a comparison of *Wolbachia* density across suburbs. In June 2019, we measured *w*Mel density in adults (15 females and 15 males) from laboratory populations that were established from field collections at different times (Gordonvale at $F_1$, Yorkeys Knob at $F_1$ and $F_{16}$, central Cairns at $F_9$ and the *w*Mel lab population). *Aedes aegypti* from field collections and laboratory populations were screened for *Wolbachia* infection status and density using a Roche Lightcycler 480 according to previously described methods [51]. Genomic DNA was extracted with 250 μL of 5% Chelex 100 Resin (Bio-Rad laboratories, Hercules CA) and 3 μL of Proteinase K (20 mg/mL) (Roche Diagnostics Australia Pty. Ltd., Castle Hill New South Wales, Australia). Tubes were incubated for 60 min at 65˚C then 10 min at 90˚C. Three primer sets were used to amplify markers specific to mosquitoes (mRpS6_F 5'AGTTGAACG-TATCGTTTCCCGCTAC3' and mRpS6_R 5' GAAGTGACGCAGCTTGTGGTCGTCC3'), *Ae. aegypti* (aRpS6_F 5'ATCAAGAAGCGCCGTGTCG3' and aRpS6_R 5'CAGGTGCAGGA TCTTCATGTATTCG3'), and *w*Mel (w1_F 5'AAAATCTTTGTGAAGAGGTGATCTGC3' and w1_R 5' GCACTGGGATGACAGGAAAAGG3'). Relative *Wolbachia* densities were determined by subtracting the crossing point (Cp) value of the *w*Mel-specific marker from the Cp value of the *Ae. aegypti*-specific marker. Differences in Cp were averaged across 3 consistent replicate runs, then transformed by $2^n$.

## Phenotypic comparisons

We performed two sets of experiments to evaluate the phenotypic effects of *w*Mel derived from field and laboratory populations. In the first set, populations were cured with tetracycline to remove the *w*Mel infection, maintaining similar genetic backgrounds between infected and uninfected lines. In the second set, we used backcrossing to introduce the *w*Mel infection from different origins to a common genetic background, then compared populations to uninfected lines that had been crossed to the same background. Both sets of experiments involved the *w*Mel Lab population which was collected from Cairns in 2014 and had spent at least 60 generations in the laboratory before the first set of experiments commenced. All experiments were performed at 26˚C and a 12:12 light:dark cycle.

Experiment 1 was performed in September 2018 using *w*Mel-infected populations collected from Yorkeys Knob in February 2018 (*w*Mel YK) and Cairns in 2014 (*w*Mel Lab). *w*Mel YK and *w*Mel Lab were divided into four population cages each. Two replicate populations from each line were treated for three consecutive generations with tetracycline hydrochloride (2 mg/mL) provided to adults in 10% sucrose solution to cure the *w*Mel infection. Females were blood fed at 10 d old to ensure that they had fed on the antibiotic solution prior to blood feeding. The other two replicate populations from each line were left untreated but reared synchronously. All populations were reared in the absence of antibiotics for two generations before experiments commenced, with thirty adults from each population screened for *Wolbachia* infection to ensure complete removal of *w*Mel in the treated lines (*w*Mel YK.tet and *w*Mel Lab. tet) and fixation of *w*Mel in the untreated lines (*w*Mel YK and *w*Mel Lab). The *w*Mel YK and *w*Mel YK.tet populations were at $F_8$ in the laboratory when experiments commenced.

Experiment 2 was performed in August 2020 using *w*Mel-infected populations collected from Yorkeys Knob in February 2020 (*w*Mel YK), Gordonvale in February 2020 (*w*Mel GV), Cairns in 2014 (*w*Mel Lab), as well as an uninfected population that had been cured of *w*Mel

in the previous experiment (wMel Lab.tet). Two hundred females from each population were crossed to 200 males from a natively uninfected population collected from locations in Cairns prior to wMel releases. This process was repeated for two further generations to produce a similar nuclear background between populations. The backcrossed wMel YK and wMel GV populations were at $F_5$ in the laboratory when experiments commenced.

In both experiments, we measured larval development time, adult wing length, female fecundity and egg hatch proportions. Eggs (<1 week old) from each population were hatched in reverse osmosis (RO) water and 100 larvae (<1 d old) were counted into plastic trays filled with 500 mL RO water (with 6 replicate trays per population in experiment 1 and 12 replicate trays per population in experiment 2). Larvae were fed TetraMin Tropical Fish Food tablets (Tetra, Melle, Germany) *ad libitum* (with daily monitoring and removal of excess food to prevent overfeeding) until pupation. Development time was scored by counting and sexing pupae twice per day. Adults were pooled across replicate trays and released into cages. Wings of adults (20 males and 20 females per population) were dissected and measured for their length (from the alular notch to the wing tip). Females (5–7 d old, sugar-starved for 1 d) were blood fed and isolated in 70 mL specimen cups filled with 15 mL of larval rearing water, lined with a strip of sandpaper (Norton Master Painters P80; Saint-Gobain Abrasives Pty. Ltd., Thomastown, Victoria, Australia) and covered with a mesh lid. Twenty females were isolated per population in experiment 1. In experiment 2, fecundity and egg hatch proportions were tracked across four consecutive gonotrophic cycles by isolating 30 engorged females per population, returning females that laid eggs to cages, then blood feeding and isolating 30 engorged females again every 4–5 d. Eggs were collected 4 days after blood feeding, partially dried, then hatched 3 d after collection in trays filled with RO water and a few grains of yeast. Fecundity and egg hatch proportions were determined by counting the total number of eggs and the number of hatched eggs (with the egg cap clearly detached) under a dissecting microscope. In experiment 2, we also measured adult longevity (8 replicate cages of 25 males and 25 females per population) by maintaining adults in 3 L cages with cups of 10% sucrose and scoring mortality 3 times per week. Quiescent egg viability was measured in experiment 2 by storing eggs on sandpaper strips in a sealed container with a saturated solution of potassium chloride to maintain a constant humidity of ~80%. Twelve replicate batches of eggs per population (median 68 eggs) were hatched on week 1 and 2, then every two weeks until week 22. Egg hatch proportions were scored as above for individual females.

## Effects of tetracycline treatment on fitness

In the above experiments we used populations that had been cleared of *Wolbachia* infections through tetracycline treatment. Despite allowing for at least two generations of recovery, these treatments could potentially disrupt the microbiome (including the mitochondria), leading to fitness differences between lines that are independent of *Wolbachia* infection. To test whether antibiotic treatment has any effect on fitness, we fed wMel-infected (wMel Lab) and natively uninfected ($F_{29}$ in the laboratory) adults with 2 mg/mL tetracycline hydrochloride for 10 d before blood feeding, then measured fitness in the subsequent generation. We scored larval development time, survival to pupa and sex ratio (6 replicate trays per treatment) as well as female and male wing length, female fecundity and egg hatch proportions (20 individuals each). Experiments were performed identically to the phenotypic comparisons above.

## Cytoplasmic incompatibility

We tested the ability of wMel-infected *Ae. aegypti* to induce cytoplasmic incompatibility in near-field and laboratory populations. Reciprocal crosses were performed in June 2019

between *w*Mel-infected populations established from Gordonvale at $F_1$ (*w*Mel GV $F_1$), Yorkeys Knob at $F_1$ (*w*Mel YK $F_1$) or Cairns at $F_{60+}$ (*w*Mel Lab), and a natively uninfected population ($F_{35}$ in the laboratory). Pupae from each population were sexed and released into separate 3 L cages to confirm accurate sex sorting. Groups of 40 females and 40 males (1 d old) were then aspirated into cages together and left for 5 d to mate. Females (starved of sugar for 24 hr) were blood-fed, isolated for oviposition and scored for fecundity and egg hatch proportions according to the methods described above for the phenotypic comparisons.

### *Wolbachia* and mitochondrial whole genome sequencing

Previously, we sequenced the *w*Mel and mitochondrial genomes of *Ae. aegypti* collected up to eight years after field releases [39]. Here, we extend these findings by sequencing the *Wolbachia* and mitochondrial genomes of *w*Mel-infected *Ae. aegypti* sampled a decade apart from pre-release and post-release populations. The pre-release *w*Mel-infected populations (*w*C45 $F_9$ and $F_{10}$) were sampled in 2010 and stored at -20°C, while three *w*Mel-infected populations were sampled in 2020: *w*Mel GV $F_2$ collected from Gordonvale in 2020, *w*Mel YK $F_2$ collected from Yorkeys Knob in 2020, and *w*Mel Lab, collected from Cairns in 2014 and maintained under laboratory conditions until sampling. Genomic DNA was extracted from pooled samples containing five adult females. Sequencing libraries were then prepared as described previously [52].

### Reference genome assembly

Sequencing reads were quality filtered using Trimmomatic [53]. The samples were trimmed in paired-end mode with the following parameter settings: leading = 20; trailing = 20; slidingwindow = 4:20; minlen = 70; adapter sequences were removed using the ILLUMINACLIP option, with maximum seed mismatches = 2 and the palindrome clip threshold = 30. Reads were aligned to a *w*Mel reference genome (GenBank accession: NC_002978.6) and an *Ae. aegypti* mitochondrial reference genome (GenBank accession: MH348177.1) with the Burrows-Wheeler Aligner (BWA; [54]) using the bwa mem algorithm and default parameter settings. Quality filtering of alignments and variant calling was performed with SAMtools and BCFtools [55,56]. PCR duplicates were excluded from subsequent analyses by soft masking. Reads with a MAPQ score < 25 were removed from the alignment, except for reads with MAPQ = 0, which were permitted to allow for mapping to repetitive regions. Genotype likelihoods were calculated using a maximum of 2000 reads per position. For variant calling, ploidy was set to haploid. The variant call output was used to create a consensus nucleotide sequence, wherein genome positions with coverage < 5 were masked as 'N'. Loci were considered to be polymorphic within a sample if they had coverage $\geq$ 30 and two or more alleles with a frequency of $\geq$ 25%. Any such loci occurring within the 16S rRNA or 23S rRNA genes were excluded from analysis, as we observed a relatively high level of contaminant reads mapping to these regions. Genome sequences were inspected and aligned with Geneious v 9.1.8 (https://www.geneious.com).

Kraken2 [57] and the Standard-8 precompiled reference database (https://benlangmead.github.io/aws-indexes/k2; downloaded 17/9/21) were used to search for sequence contamination within the *Wolbachia* genomes. The sequencing reads mapped by bwa to the *w*Mel reference genome were filtered to remove reads matching taxa other than *Wolbachia*, and genome assemblies were then repeated with the filtered datasets, using the above pipeline. The original pre-filtration genome sequences were edited to correct erroneous positions after comparison with the corresponding post-filtration genome sequences.

## Statistical analysis

Most experimental data (including *Wolbachia* density, development time, wing length, fecundity and egg hatch) were analyzed with general linear models (GLMs) while adult longevity data were analysed with log-rank tests. All analyses were carried out with IBM SPSS Statistics 26. Data were transformed where appropriate (with all proportional data being logit transformed). The first experiment had two replicate populations and data were initially analysed with replicate population (nested within population origin x *Wolbachia* infection status) included as a factor. Replicate populations were then pooled for a second analysis due to a lack of significant effect of replicate population (P > 0.1) for any trait. *Wolbachia* infection status and population origin were included as factors. In the second experiment, fecundity and egg hatch proportions were tracked across gonotrophic cycles with the same mosquitoes, so we ran separate analyses for each cycle. For quiescent egg viability, *w*Mel-infected and uninfected populations were analysed both together and separately at the first (1 week) and last time (22 weeks) points. Field *Wolbachia* density data were analysed with release year and suburb (nested within release year) as factors. We performed Bonferroni corrections where multiple traits were evaluated in the same cohort of mosquitoes.

## Supporting information

**S1 Fig. *w*Mel *Wolbachia* infection frequencies in Cairns in (A) 2016 and (B) 2018 sampled through ovitrapping.** Contains information from OpenStreetMap and OpenStreetMap Foundation, which is made available under the Open Database License.
(TIF)

**S1 Table. Observed differences in *Wolbachia* and mitochondrial genomes.** Positions are shown relative to the GenBank reference sequences used for read mapping. Some or all of the differences are likely to be sequencing artifacts (see main text). For comparison, the results of Huang et al. [39] and Dainty et al. [40] are also shown. ND = not determined.
(XLSX)

**S2 Table. Genome positions found to show polymorphism within populations.** In our study, loci were considered to be polymorphic within a sample if they had coverage $\geq$ 30 and two or more alleles with a frequency of $\geq$ 25%. Any such loci occurring within the 16S rRNA or 23S rRNA genes were excluded from analysis, as we observed a relatively high level of contaminant reads mapping to these regions. Positions are shown relative to the GenBank reference sequences used for read mapping. For comparison, the results of Huang et al. [39] and Dainty et al. [40] are also shown. For Dainty et al. [40], multiple samples per population were analysed, with each sample containing a single individual—allele frequencies are reported as number individuals/total individuals per population. NR = not reported; not scored as polymorphic by the authors of the original study; in most cases these values are likely to be at or close to 100%.
(XLSX)

## Acknowledgments

We thank Safi Soleimannejad and Kelly Richardson for technical assistance.

## Author Contributions

**Conceptualization:** Perran A. Ross, Ary A. Hoffmann.

**Data curation:** Katie L. Robinson.

**Formal analysis:** Perran A. Ross, Katie L. Robinson, Qiong Yang, Ashley G. Callahan, Ary A. Hoffmann.

**Funding acquisition:** Ary A. Hoffmann.

**Investigation:** Perran A. Ross, Katie L. Robinson, Qiong Yang, Ashley G. Callahan, Thomas L. Schmidt, Jason K. Axford, Marianne P. Coquilleau, Kyran M. Staunton, Michael Townsend, Scott A. Ritchie, Xinyue Gu.

**Methodology:** Perran A. Ross, Katie L. Robinson, Qiong Yang.

**Resources:** Scott A. Ritchie, Meng-Jia Lau, Ary A. Hoffmann.

**Supervision:** Ary A. Hoffmann.

**Visualization:** Perran A. Ross.

**Writing – original draft:** Perran A. Ross, Katie L. Robinson, Ary A. Hoffmann.

**Writing – review & editing:** Perran A. Ross, Katie L. Robinson, Qiong Yang, Ashley G. Callahan, Thomas L. Schmidt, Jason K. Axford, Marianne P. Coquilleau, Kyran M. Staunton, Michael Townsend, Scott A. Ritchie, Meng-Jia Lau, Xinyue Gu, Ary A. Hoffmann.

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
