## [Decision Letter · Decision Letter 0]

17 Dec 2021

Dear Dr. Ross,

Thank you very much for submitting your manuscript "A decade of stability for wMel Wolbachia in natural Aedes aegypti populations" for consideration at PLOS Pathogens. As with all papers reviewed by the journal, your manuscript was reviewed by members of the editorial board and by several independent reviewers. The reviewers appreciated the attention to an important topic. Based on the reviews, we are likely to accept this manuscript for publication, providing that you modify the manuscript according to the review recommendations.

Sincerely,

George Dimopoulos, PhD MBA

Guest Editor

PLOS Pathogens

Raul Andino

Section Editor

PLOS Pathogens

Kasturi Haldar

Editor-in-Chief

PLOS Pathogens

orcid.org/0000-0001-5065-158X

Michael Malim

Editor-in-Chief

PLOS Pathogens

orcid.org/0000-0002-7699-2064

Reviewer Comments (if any, and for reference):

Reviewer's Responses to Questions

**Part I - Summary**

Reviewer #1: The manuscript entitled "A decade of stability for wMel Wolbachia in natural Aedes aegypti populations" brings relevant new information regarding the use of Wolbachia as an agent to mitigate arbovirus transmission. Although Wolbachia releases as a replacement strategy is ongoing in several countries worldwide, data on Cairns are of particular interest because it was the first place to accomplish invasion into wild Aedes aegypti populations.

Reviewer #2: This is a key descriptive study for understanding the field stability of the wMel infection in Ae. aegypti, given that the Australian releases were the first completed. It offers a comparator for other more recent field release sites, some of which have struggled to exhibit the successful spread of Wolbachia. They demonstrate few fitness effects and little change in the wMel:Ae. aegypti relationship over time. They also demonstrate few genetic changes in either the Wolbachia genome or the mosquito mitochondrial genome.

The fitness measures in mosquitoes carried out here are bread and butter for this lab. They have therefore been well designed, executed, analyzed statistically, and portrayed graphically. The same is true for the Wolbachia density measures and CI test crosses.

The authors have been careful not to over-interpret genetic changes they have found that are likely due to sequencing errors.

The article is well written.

**Part II – Major Issues: Key Experiments Required for Acceptance**

Reviewer #1: The manuscript reports common garden methodologies to investigate fitness cost of Wolbachia considering different mosquito genetic backgrounds, as well as other protocol such as Wolbachia frequency and density in mosquitoes, cytoplasmic incompatibility and Wolbachia/mitochondria genome sequencing.

Conclusions are supported by data.

Reviewer #2: None

**Part III – Minor Issues: Editorial and Data Presentation Modifications**

Reviewer #1: Although the manuscript provide new and relevant information, some additional details are required in specific sections of the manuscript.

Line 77. Authors mention that in some locations, Wolbachia infection has remained at an intermediate frequency or dropped out, requiring supplemental releases. For sure, environmental, entomological and likely human behavior aspects influence this outcome. I'd recommend authors to add a few sentences in the Discussion section correlating how their lab-based results could help explaining (at least partially) the reason why some areas would still need supplemental releases to achieve a stable invasion.

Line 178: In the phenotypic comparison assays, authors observed fecundity and egg hatch proportions during four consecutive gonotrophic cycle and compared data using GLMs. One of the most important assumptions of Generalized linear models is the independence among data points. From a personal perspective, I wouldn't say the number of eggs from a given gonotrophic cycle is fully independent from the previous one, especially when mosquito females are kept in cages.

Line 356: On this manuscript, authors report the spatiotemporal variation of Wolbachia density in 4th instar larvae hatched from ovitraps collected across Cairns suburbs in February-March 2018. However, the research group leading this manuscript has data from older field samplings. Would still be possible to evaluate potential changes/fluctuations of Wolbachia density over this 10 years period? Or considering data already published, the use of different primers make a direct comparison not recommended?

Line 398: What does it mean? Adding food ad libitum would make the water cloudy, strongly affecting rearing conditions and thus larval mortality.

Line 421: Authors claim they allowed multiple generations of recovery from tetracycline treatment before start the experiments. But I understood it was for only two generations.

Reviewer #2: I think the ‘Discussion’ could use an additional speculative section on why other places ‘don’t look like Cairns’. Why if this symbiont has so little effect on populations is it falling out of Vietnamese and Brazilian populations? What is known about fitness in these other regions?

PLOS authors have the option to publish the peer review history of their article (what does this mean?). If published, this will include your full peer review and any attached files.

Reviewer #1: No

Reviewer #2: No

Figure Files:

Data Requirements:

Reproducibility:

References:

---

## [Editor Report · Decision Letter 1]

7 Jan 2022

Dear Dr. Ross,

We are pleased to inform you that your manuscript 'A decade of stability for wMel Wolbachia in natural Aedes aegypti populations' has been provisionally accepted for publication in PLOS Pathogens.

Best regards,

George Dimopoulos, PhD MBA

Guest Editor

PLOS Pathogens

Raul Andino

Section Editor

PLOS Pathogens

Kasturi Haldar

Editor-in-Chief

PLOS Pathogens

orcid.org/0000-0001-5065-158X

Michael Malim

Editor-in-Chief

PLOS Pathogens

orcid.org/0000-0002-7699-2064
---

## [Editor Report · Acceptance letter]

17 Feb 2022

Dear Dr. Ross,

We are delighted to inform you that your manuscript, "A decade of stability for wMel Wolbachia in natural Aedes aegypti populations," has been formally accepted for publication in PLOS Pathogens.

Best regards,

Kasturi Haldar

Editor-in-Chief

PLOS Pathogens

orcid.org/0000-0001-5065-158X

Michael Malim

Editor-in-Chief

PLOS Pathogens

orcid.org/0000-0002-7699-2064